# Development of an Antigen-Antibody Co-Display System for Detecting Interaction of G-Protein-Coupled Receptors and Single-Chain Variable Fragments

**DOI:** 10.3390/ijms22094711

**Published:** 2021-04-29

**Authors:** Yinjie Zhang, Boyang Jason Wu, Xiaolan Yu, Ping Luo, Hao Ye, Yan Yu, Wei Han, Jingjing Li

**Affiliations:** 1Laboratory of Regeneromics, School of Pharmacy, Shanghai Jiao Tong University, Shanghai 200240, China; zhangyinjie88@sjtu.edu.cn (Y.Z.); 13262631086@163.com (P.L.); yehao715@126.com (H.Y.); 2Department of Pharmaceutical Sciences, College of Pharmacy and Pharmaceutical Sciences, Washington State University, Spokane, WA 99210, USA; boyang.wu@wsu.edu; 3Shanghai Municipality Key Laboratory of Veterinary Biotechnology, School of Agriculture and Biology, Shanghai Jiao Tong University, Shanghai 200240, China; xiaolanyu89@163.com

**Keywords:** yeast two hybrid, G protein coupled receptor, antibodies, off-targets, epitope

## Abstract

G-protein-coupled receptors (GPCRs), especially chemokine receptors, are ideal targets for monoclonal antibody drugs. Considering the special multi-pass transmembrane structure of GPCR, it is often a laborious job to obtain antibody information about off-targets and epitopes on antigens. To accelerate the process, a rapid and simple method needs to be developed. The split-ubiquitin-based yeast two hybrid system (YTH) was used as a blue script for a new method. By fusing with transmembrane peptides, scFv antibodies were designed to be anchored on the cytomembrane, where the GPCR was co-displayed as well. The coupled split-ubiquitin system transformed the scFv-GPCR interaction signal into the expression of reporter genes. By optimizing the topological structure of scFv fusion protein and key elements, including signal peptides, transmembrane peptides, and flexible linkers, a system named Antigen-Antibody Co-Display (AACD) was established, which rapidly detected the interactions between antibodies and their target GPCRs, CXCR4 and CXCR5, while also determining the off-target antibodies and antibody-associated epitopes. The AACD system can rapidly determine the association between GPCRs and their candidate antibodies and shorten the research period for off-target detection and epitope identification. This system should improve the process of GPCR antibody development and provide a new strategy for GPCRs antibody screening.

## 1. Introduction

Monoclonal antibody drugs are highly favored due to their excellent target selectivity and long half-life [1,2,3]. One of the primary targets of antibody drugs are membrane proteins, especially multi-pass membrane proteins such as G-protein-coupled receptors (GPCRs) and ion channel proteins. They play essential roles in the cellular function and are prime targets for disease treatment and diagnosis, the main use of monoclonal antibody drugs. However, according to marketed antibody drug statistics in 2019, only two antibody drugs target GPCRs; the others are all single transmembrane protein or adhesion molecule antibodies that do not target ion channel or transporter proteins [4,5,6]. This reflects the difficulty of developing GPCRs antibodies.

Antibody development for GPCRs is challenging because both the hybridoma technique and antibody library display technique require proteins with natural structures as antigens. Earlier studies used intact membrane proteins as immunogens and purified antigens using agents such as detergents [7]. Therapeutic antibodies require recognition of the natural conformation of antigens, and it is exceedingly difficult to obtain membrane proteins with natural conformation due to the complexity of reconstitution of purified proteins with phospholipid microcapsules. Because multi-pass membrane proteins have several discontinuous extracellular regions, transmembrane regions, and intracellular regions, they interact and influence each other. When a particular segment of the extracellular region is expressed alone, its conformation will inevitably create unexpected changes in the natural structure, and the prepared antibodies cannot recognize the protein.

The development of the yeast ubiquitin system provides a new vision for the present dilemma [8]. The system expresses two receptor membrane proteins via fusion transmembrane peptides on the yeast cell membrane surface. The receptor modules are conjugated in the Cub and Nub of the ubiquitin detection system, and when the receptors interact, the detection system is activated by releasing transcription factors and turning on the reporter gene.

Inspired by this study, we developed the Antigen-Antibody Co-Display system (AACD). The AACD system integrates natural membrane protein display technology and antibody display technology, where protein-protein interactions (PPI) between membrane proteins and a single-chain variable fragment (scFv) displayed on the yeast cell surface are detected by the split-ubiquitin detection system.

AACD offers several advantages: (1) antigens can be presented in their native form on the cell surface; (2) large-scale production of the antigen of interest is unnecessary; and (3) it is unnecessary to produce large-scale antibodies for specificity and antigen epitope assays.

The membrane protein CXCR4 and a patented scFv [9] of CXCR4 were chosen as a model to verify the feasibility of the system. The signal peptide, transmembrane peptide, and flexible linker were optimized to increase system robustness.

## 2. Results

### 2.1. Design of the AACD System 

A human GPCR, CXCR4, and its antibody encoding gene were used to test the effect of the AACD system on membrane antigen and scFv interaction. The structure of the antigen modules (Figure 1A) was composed of three parts, including a signal peptide, CXCR4 as the antigen, and a reporter unit. The signal peptide was optimized by testing the signal peptide of different genes, including *S. cerevisiae*-secreted *Mfal1* or human *CXCR4*. The reporter module consisted of the *C*-terminus of ubiquitin (C_ub_) and a transcription factor, GAL4. The sketch map of the antibody module is shown in Figure 1B. A scFv was located in the periplasm and anchored onto the plasma membrane via a linker and a transmembrane peptide, followed by a reporter unit, a mutant of the N-terminus of ubiquitin (N_ub_*G*) located in the cytoplasm. The N_ub_*G* and C_ub_ domains constituted a split-ubiquitin system, in which the I13G mutation in the N_ub_ domain blocked the spontaneous association of the two domains.

The detection of antigen-antibody interaction is shown in Figure 1B. Once the scFv was bound to GPCR, the split-ubiquitin was reconstructed to a whole and activated ubiquitin, which was recognized by ubiquitin-specific protease (UBP), leading to the release of GAL4 from the C-terminus of C_ub_. GAL4, was guided into the nucleus by an SV40 nucleus localization signal and initiated the expression of reporter genes *HIS3, ADE2*, and *LacZ*, causing cell survival and colorization on the selective plate. N_ub_*I*, the wild-type ubiquitin N-terminus, was used as positive control, which could associate with C_ub_ spontaneously, leading to the release of GAL4 independent of the antigen-antibody interaction.

### 2.2. Topological Structure of Antibody Modules

The antibody module was expressed via two topological structures, *trans* and *cis* models (Figure 2A). In the *trans*-model, N_ub_*G* was expressed at the *N*-terminal (NT). The scFv was secreted outside the cell by a transmembrane peptide, forming a type II transmembrane protein. In the *cis*-model, scFv was expressed at the NT of the reading frame, secreted by the signaling peptide upstream, connected with N_ub_*G* in the cell via a transmembrane peptide, forming a type I transmembrane protein. The difference between two protein topological structures was the orientation of the linker peptide of scFv and the transmembrane protein. The *trans*-structured linker peptide was connected to the N-terminus of scFv, while the *cis*-structured linker was connected to the C-terminus of scFv.

To test and optimize the system, we used a patented monoclonal CXCR4 antibody. The variable region of the heavy-chain (VH) and the variable region of the light-chain (VL) were recombined into scFv with 3xGGGGS flexible peptides between them, named At1-scFv [9]. The transmembrane structures in the two topological models were predicted by TMHMM Server v. 2.0 software (http://www.cbs.dtu.dk/services/TMHMM/, accessed on 14 May 2019) [10] (Appendix A). The interaction of CXCR4 and At1-scFv was used to test the system, and the following tests were also performed this way.

The efficiency of antibody modules with two topological structures was evaluated by detecting their interaction with the CXCR4 antigen (Figure 2B,C). The results showed that the yeast with antibody module of trans-topological structure and CXCR4-C_ub_-GAL4 grew better in the selective plate and had higher β-Gal activity than the cells with the cis-topology. It demonstrated that the trans-topology significantly activated the reporter gene, while the cis structure did not. It suggests that the length of the flexible peptide might hinder the orientation of the cis-structured scFv antigen-binding region toward the cell membrane, and therefore the trans-topology might be more suitable for the detection of multiple transmembrane proteins such as CXCR4.

### 2.3. Optimization of Transmembrane Peptides

The transmembrane peptide is the most crucial element in antibody modules [11]. Three transmembrane peptides from three transmembrane proteins were utilized in the antibody module (Figure 3A) and tested for their mediated localization efficiency and interaction strength with the antigen. The three peptides were derived from the yeast Alg5 (2~25aa), named TM_alg_, a transmembrane peptide (533~565aa) of yeast protein Pho_90_, named TM_pho_, and a transmembrane peptide of the mouse CXCR5 (162~194aa), named TM_r54_. The results showed that all three transmembrane peptides were correctly located on the yeast cell membrane (Figure 3D), and the PPI between CXCR4 and At1-scFv meditated by these peptides was observed in growth (Figure 3C) and β-Gal activity assays (Figure 3B). TM_pho_ was selected as the optimal transmembrane peptide for the following experiments as it induced the strongest interaction signal.

### 2.4. Optimization of Flexible Peptide Length

In this study, a 3×GGGGS flexible peptide was used as a linker between the scFv and the transmembrane peptide. The scFv module with 0×, 3×, 6×, or 10×GGGGS was tested for PPI strength with the antigen module to optimize the flexible peptide (Figure 4). The results showed that except for the antibody without a linked GGGGS, which had no reporter gene expression, the reporter gene intensity of the three lengths of flexible peptides was the same. The flexible peptide with the shortest length (3×GGGGS) was selected for the following experiments to avoid unspecific binding between the flexible linker and the antibody module.

### 2.5. The Effects of Signal Peptides of the Antigen Protein

The antigen used in this study was CXCR4, a seven-transmembrane protein. CXCR4 does not have an N-terminal signal peptide, but its transmembrane peptide, as a localization signal, locates it to the cell membrane. In the yeast cells, we confirmed whether human CXCR4 could be correctly anchored to the cell membrane. Generally, the α-factor signal peptide is the most commonly used yeast-based protein expression system and yeast surface display technique [12], which could guide the secretion of the target protein into the periplasm. Therefore, CXCR4 only and CXCR4 with an α-factor signal peptide fused at the N-terminus were compared for their localization and PPI strength with At1-scFv. The results showed that both the CXCR4 signal peptide and the α-factor signal peptide could locate the antigen module to the cell membrane. The CXCR4 without α-signal peptides produced better reporter signals (Figure 5).

### 2.6. Antibody Specificity Detection

Six scFv genes were selected to evaluate the selection specificity of the AACD system. They included two patented CXCR4 antibodies (At1 [9], At2 [13]), one CXCR5 antibody (At3 [14]), and three unrelated antibodies maintained by our lab. The antigen modules of the system were combined with human-CXCR4 and human-CXCR5 to conduct hybridization experiments. The results of flow cytometry analysis showed that At1-IgG could interact with CXCR4 but not CXCR5, whereas At2-IgG and At3-IgG could interact with both CXCR4 and CXCR5 (Figure 6B). This revealed that At2 and At3 could cross react with CXCR4 and CXCR5, while At1 did not. The same results tested using the AACD system are shown in Figure 6A. These results suggest that the AACD system could correctly detect the cross-reactions between the antibody and antigens.

### 2.7. Epitope Assay of CXCR4

To analyze the antigen epitope that mediated the At1-scFv and CXCR4 interaction, four extracellular fragments of CXCR4, including an NT fragment and three extracellular loops (ECLs), were exchanged with the flexible peptide 2×GGGGS. The interactions of CXCR4_ΔECL1_ and CXCR4_ΔECL2_ with the antibody were blocked, while those of CXCR4_ΔNT_ and CXCR4_ΔECL3_ with the antibody were almost the same as the wild-type CXCR4 antigen (Figure 7B). The results suggested that At1-scFv was against a discontinued epitope on CXCR4 composed of ECL1 and ECL2.

## 3. Discussion

Since the critical role of GPCRs in the regulation of physiological functions was first recognized, they have become the main target of antibody drug development. The main challenge in developing antibodies against GPCRs is their unique structure. The GPCR has the common structural feature of seven-transmembrane peptides: the conformation of a barrel-shaped transmembrane protein in which the extracellular part is composed of four discontinued fragments, including an NT domain and three ECLs. The four fragments form a stable structure via disulphide bonds and non-covalent bonds. On the one hand, the expression of any fragment alone likely results in the loss of its natural conformation, while the full-length proteins with natural conformation are hard to purify. GPCRs are strictly dependent on cytomembrane and difficult to operate in vitro. Operations on mammalian cells are unusually time-consuming and laborious. However, as a eukaryote, yeast cells can well express mammalian proteins with correct conformation, and their short generation cycles and easy accessibility allow assays to be finished within a short time.

The AACD Y2H system can rapidly detect the interaction between an antibody and a GPCR without antigen expression and purification. The system displayed GPCR antigens in situ on the cytomembrane in their natural conformation. Moreover, the antibodies were also displayed on the membrane surface. The split-ubiquitin reporter system, coupled to the antigen and antibody, made it possible to visualize the strength of antigen-antibody interactions. 

To optimize the system, we compared the effects of different topological structures, transmembrane peptides, and flexible linkers of antibody modules. The effectiveness of *trans* or *cis* topological structures was compared. The two structures were both composed of an extracellular scFv and an intracellular N_ub_*G*, the difference being that in the trans-structure the linker was associated with NT of scFv, which possibly introduced steric hindrance and blocked antigen-antibody interactions. In the cis-structure, the linker was coupled with the scFv C-terminus, where scFv might have to undergo larger, potentially unfavorable conformational changes paired with the problem of steric hinderance when used in this topology. As proved by the results, only the trans-structure could establish the interaction between the scFv and CXCR4. Therefore, it was determined as the classic structure of the antibody module.

The transmembrane peptide is an essential element in the platform. A transmembrane peptide not only correctly anchors the antibody outside the cell membrane but also minimally affects the expression or the structure of scFv without causing non-specific binding. To test this idea, we used transmembrane peptides of different transmembrane proteins to observe their subcellular localization and reporter signal strength. TM_pho_ showed a stronger reporter signal. Therefore, it was selected as the optimal transmembrane peptide for subsequent experiments. 

In protein engineering, flexible peptides are often added between the subunits of fusion proteins to make these subunits oscillate and rotate freely, reducing mutual steric hindrances and keeping their expected biological functions [15]. Here, we explored the topological structure of the antibody and speculated that the flexible peptide might affect the spatial binding of the antigen to the antibody. In general, too-short flexible peptides cause steric hindrance, preventing proteins from approaching each other, while the too-long flexible peptides not only cause non-specific binding but also the intertwining of extra flexible peptides [16]. To select an appropriate length of flexible peptide and observe its influence on the interaction, we tested three flexible peptides of different lengths in the antibody module. The results showed that the antibodies containing three lengths of flexible peptides exhibited the same interaction strength with the antigen (Figure 4). Thus, the flexible peptide with the shortest length, 3×GGGGS, was selected in the same strength of PPIs. 

With the AACD system, many assays of GPCR-antibody interactions are anticipated to become much easier. First, potential off-target candidate antibodies could be rapidly detected. With high sequence homology, the off-targets of GPCR antibodies likely happen and probably produce side effects in drug use. The AACD system provides an excellent platform for rapid scanning of possible off-targets. For example, in this study the cross-reactions between antibodies and CXCR4 or CXCR5 were assayed, demonstrating an impressively convenient and straightforward process. In practice, to extensively detect the cross-reactions, the more advanced high-throughput assays, such as those including the whole chemokine receptor family, could be done easily as well, though the parameters of the binding kinetics still need to be measured by other methods.

Second, the epitopes on GPCRs that mediate the binding of antibodies could be determined more easily. For a monoclonal antibody drug, the epitopes on the antigen that mediate antibody-antigen binding are vital [12], essential for elucidating the mechanism of variant antibody drugs. In this study, several mutants of CXCR4 were generated by deleting the N-terminus or substituting ECLs with flexible peptides, and the antibody At1-scFv interactions detected. Our results showed that the substitutions of ECL1 or ECL2 but not the NT domain or ECL3 completely halted the association of At1-scFv with CXCR4, implying that the ECL1 and ECL2 of CXCR4 might cooperatively constitute the epitope participating in the interaction with At1-scFv. The structural analysis of CXCR4 revealed that the two fragments were very close and linked to each other by a disulphide bond [17]. Although another possibility is that the ECL1 and ECL2 mutants might result in misfolding of CXCR4, our results clearly indicate the capability of the AACD system to facilitate GPCR epitope scanning.

Last but not least, the system has the potential to be developed into an antibody screening platform for GPCR targets in combination with an antibody library. The library could be an scFv library with the advantage of simple structure and easy genetic operation. However, scFv is not an ideal structure to perform antibody screening in the AACD system, since the capacity of an scFv library (10^7^~10^8^) cannot reach a high level as in yeast cells due to lower transformation efficiency [18], compared with a phage scFv library (10^10^~10^11^). An antibody library with higher capacity, such as the Fab library, should be employed to make the system competitive. By mating two yeast strains harboring a library of VH and VL respectively, the capacity of the Fab library is sufficient for antibody scanning, which is expected to exceed 10^9^ according to the methods used by Weaver-Feldhaus et al. [19] and Baek et al. [20]. In addition, the conformation of the antigen-binding site of Fab is closer to that of IgG than scFv, which decreases the risk of affinity loss after being restored to the IgG form [21]. Therefore, we will focus on applying the Fab library in the AACD system in future studies.

## 4. Materials and Methods

### 4.1. Strains and Plasmids 

The plasmid construction and amplification host was the *E. coli* strain DH5α. The antigen and scFv were constructed and displayed in AH109 (trp1-901, leu2-3, 112, ura3-52, his3-200, gal4Δ, gal80Δ, LYS2::GAL1_UAS_-GAL1_TATA_-HIS3, GAL2_UAS_-GAL2_TATA_-ADE2, URA3::MEL1_UAS_-MEL1_TATA_-lacZ, MEL1) and Y187 (MATα, ura3-52, his3-200, ade2-101, trp1-901, leu2-3, 112, gal4Δ, met-, gal80Δ, URA3::GAL1_UAS_-GAL1_TATA_-lacZ) yeast cells. pGAD-T7 and pGBK-T7 plasmids (Code No. 630442, 630443), *E. coli* strain DH5α (Code No. 9057) and yeast strains (Code No. 630457, 630498) were purchased from TaKaRa, Japan. The pcDNA3.4 plasmid (Code No. A14697) and BHK21 and CHO cell strains (Code No. R76007, A37785) were purchased from Life Technologies, Waltham, MA, USA. The cells were tested for mycoplasma contamination before they were used.

### 4.2. Plasmid Construction

#### 4.2.1. Antigen Plasmids

The antigen module was expressed in the pGAD-T7 plasmid, in which all elements between the ADH1 promoter and ADH1 terminator were deleted and replaced by the target sequences using a one-step clone kit from Vazyme (one-step clone kit, Code No. C112-01, Vazyme, Nanjing, China). 

#### 4.2.2. Antibody Plasmids

The open reading frame for the antibody module was cloned into the pGBK-T7 vector, located between the ADH1 promoter and terminator. The antibody module consisted of five parts: (1) a signal peptide, (2) an scFv, (3) a linker, (4) a transmembrane peptide, and (5) a reporter unit. Each part was optimized. The signal peptide was derived from *S. cerevisiae*-secreted *MFAL1* (residues 1–89). The linker consisted of a variated number of tandem flexible peptide GGGGS. The transmembrane peptide was derived from the transmembrane peptide of *S. cerevisiae Pho90* residues 1576–1707, *ALG5* residues 4–66, or human *CXCR5* residues 463–597. The reporter unit consisted of a mutant N-terminus of ubiquitin (N_ub_*G*) for PPI detection or a GFP for localization assay.

### 4.3. Membrane Staining and Co-Localization Assay 

Yeast cells (OD_600_ = 2.0) were fixed using paraformaldehyde on ice for 10 min, then washed twice and resuspended in 0.5 mL of sterile water with 10 μM DiI (C1036, Beyotime, Shanghai, China). After incubation for 10 min, cells were washed twice. The Leica confocal system (Leica confocal SP5, Wetzlar, Germany) was used to visualize the cells (EGFP, λ_ex_: 543 nm; Dil, λ_em_: 565 nm). Three visual fields were observed for each sample.

### 4.4. Yeast Plasmid Transformation and Mating

Yeast cells were cultured in yeast peptone dextrose (YPD) medium. The antibody and antigen plasmids were transformed into Y187 and AH109 yeast cells using the LiAc method following the manufacturer’s instructions (630439, Takara, Shiga, Japan). After culture for 3 days, positive clones were selected using synthetic dropout nutrient medium/plate (SD medium) without leucine or tryptophan (SD/Leu^−^ or SD/Trp^−^). Y187 and AH109 yeast cells were mated by coculturing two strains in YPD medium overnight, shaking at 150 rpm. The hybrid strains were selected with SD/Leu^−^Trp^−^ medium.

### 4.5. Interaction Strength Detection

#### 4.5.1. Growth Assay

The heterozygotes, which were selected using SD/Leu^−^Trp^−^ medium, were plated in SD/Leu^−^Trp^−^His^−^Ade^−^ medium as 10-fold diluted cell suspensions (OD_600_ = 1, 0.1, 0.01 and 0.001 series). Images of cell growth on the plates were recorded after incubation for 3 days.

#### 4.5.2. β-Galactosidase (β-Gal) Activity Assay

The hybrid cells (OD_600_ = 0.8, SD/Leu^−^Trp^−^ medium) were lysed in 100 mL of Z-buffer (60 mM Na_2_HPO_4_, 40 mM NaH_2_PO_4_, and 10 mM KCl, 1 mM MgSO_4_, pH 7.0) through three freezing and thawing cycles in liquid nitrogen and water bath at 37 ℃. Then the cell lysates were suspended in 700 mL of Z-buffer containing 0.27% (*v*/*v*) 2-mercaptoethanol and 0.4% 2-nitrophenyl-β-D-galactopyranoside at 30 °C for chromogenic reaction. The reaction was terminated after 30~60 min by adding 400 μL of 1 M Na_2_CO_3_. The absorbance of supernatant was measured at 420 nm (OD_420_). The β-Gal activity was calculated using the following equation:$$\beta -\mathrm{galactosidase}\;\mathrm{units}=\frac{1000\times {\mathrm{OD}}_{420}}{t\left(\min\right)\times {\mathrm{OD}}_{600}}$$

OD_600_ is the absorbance of yeast culture before lysis. Each sample was tested 3 times. The *t* test was used as the statistical method.

### 4.6. Expression of Antibodies and Antigen Display

#### 4.6.1. Antibody Preparation

The heavy-chain and light-chain antibody sequences were cloned into pcDNA3.4. Then the CHO cells were transfected with the two plasmids for transient expression guided by the ExpiCHO™ Expression System (A29133, GIBCO, Waltham, MA, USA). Antibodies were purified from the supernatant of the CHO cell culture using protein G affinity chromatography (1004D, Invitrogen, Waltham, MA, USA).

#### 4.6.2. Antigen Display

The ORFs of human-*CXCR4* and human-*CXCR5* were fused with EGFP and cloned into pIRES. The BHK-21 cells were then transfected with pIRES using the Lipofectamine™ 2000 Transfection Reagent (11668027, Thermo Fisher Scientific, Waltham, MA, USA). Forty-eight hours after transfection, 2 μg/mL puromycin was added into medium, after 7 days culturing a cell pool that expressed CXCR4 or CXCR5 was generated and used for flow cytometry.

#### 4.6.3. Flow Cytometry

The BHK21 cells (3~5 × 10^6^) were resuspended in 200 mL of wash buffer (PBS containing 1.5% BSA), to which 0.5 μg of the primary antibody was added. The cells were incubated at 25 °C for 30 min, then washed three times with 1 mL of wash buffer, and incubated on ice for 30 min with the secondary detection antibody (17-4210-82, Rat anti-Mouse IgG2a Secondary Antibody, APC, eBioscience™, Waltham, MA, USA). The cells were washed twice with 1 mL wash buffer before flow cytometric analysis.

## Figures and Tables

**Figure 1 ijms-22-04711-f001:**
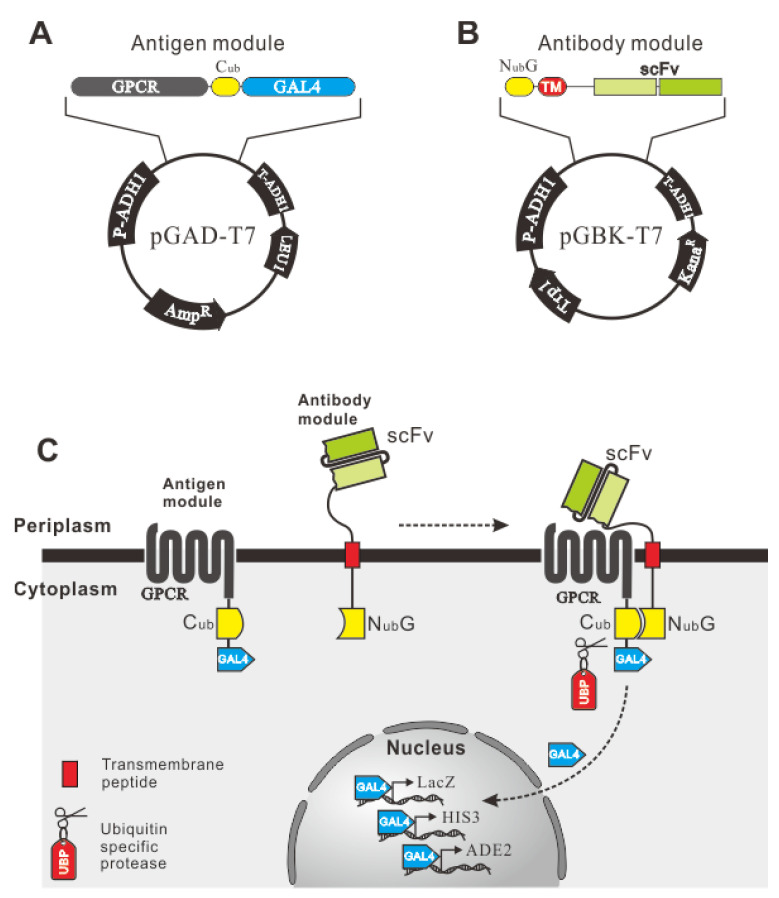
Design of the antibody and antigen plasmids and the principles of detecting the interaction. (**A**) The structure of the antigen plasmid. The antigen module is composed of G-protein-coupled receptor (GPCR), C_ub_, and GAL4 in tandem. The antigen genes were cloned into pGAD-T7. (**B**) The antibody plasmid structure. scFv is fused with a signal peptide (SP) at the N-terminus and a transmembrane peptide (TMP) at the C-terminus, followed by N_ub_*G* in tandem. The genes of the antibody module were cloned into pGBK-T7. (**C**) Principles of detecting the interaction. scFv is secreted outside the cytomembrane via SP and is linked with N_ub_*G* in the cytoplasm via the TMP. GPCR, as a transmembrane protein, is co-expressed on the plasma membrane linked to C_ub_ and GAL4 in the cytoplasm. The interaction of scFv and GPCR units C_ub_ and N_ub_*G* leads to reconstitution of split-ubiquitin, which is recognized and cleaved by UBP to release GAL4. The liberated GAL4 is transported into the nucleus via the SV40 nuclear localization signal at its N-terminus, and the transcription of reporter genes *lacZ*, *HIS3*, and *ADE2* results in blue clones on an SD/His^−^Ade^−^ plate in the presence of X-α-Gal.

**Figure 2 ijms-22-04711-f002:**
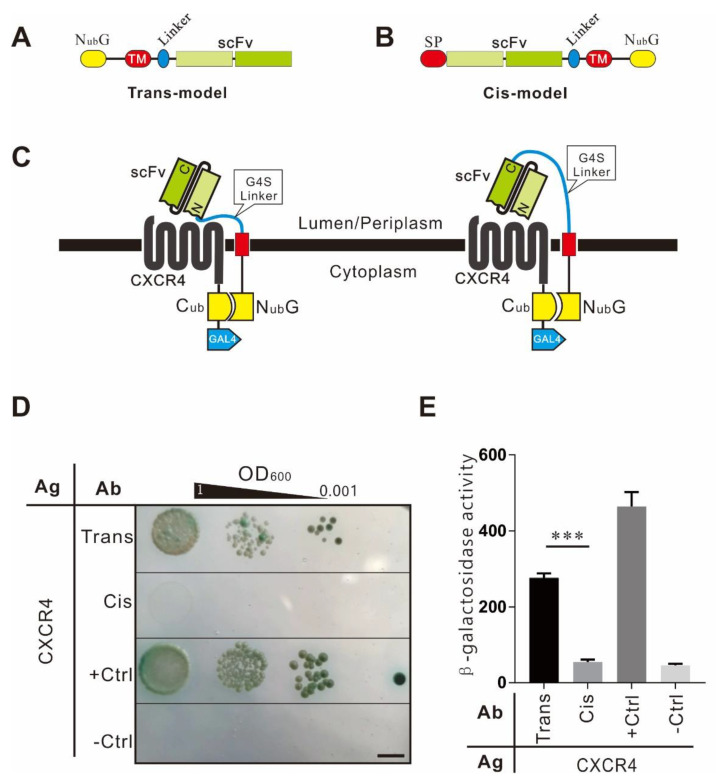
Optimization of antibody modules with different topological structures. (**A**) The *trans*-structure. The N_ub_*G* domain is fused with a type II transmembrane peptide (the C-terminus is outside of cytomembrane and the N-terminus is in the cytoplasm), followed by a linker and an scFv in tandem. (**B**) The cis structure. The scFv is fused with an α signal peptide at its N-terminus and a type I transmembrane peptide at the C-terminus (the N-terminus is secreted out of cytomembrane, and the C-terminus remains in the cytoplasm), followed by N_ub_G in tandem. (**C**) Diagram of interactions between G-protein-coupled receptor (GPCR) and scFv with two types of topological structures. (**D**) Cell growth assay. AH109 cells expressing the antigen module CXCR4-C_ub_-GAL4 and Y187 cells displaying the scFv module with the trans or cis topological structure were hybridized and plated onto SD/Leu^−^Trp^−^Ade^−^His^−^ plates containing X-α-gal. The growth of diploid cells was observed after 72 h cultivation. CXCR4-C_ub_-GAL4 × TM_pho_-N_ub_G was used as a negative control, and CXCR4-C_ub_-GAL4 × TM_pho_-N_ub_*I* was used as a positive control. Ag: antigen. Ab: antibody, the bar indicates 5 mm. (**E**) Quantitative β-Gal assay of diploid cells expressing CXCR4-C_ub_-GAL4 and two types of topological structure fusion proteins. The positive and negative controls were the same as in Figure 2D. The experiment was repeated three times independently, and every sample was tested in duplicate in each experiment (*n* = 2), *** indicates *p*-value < 0.001.

**Figure 3 ijms-22-04711-f003:**
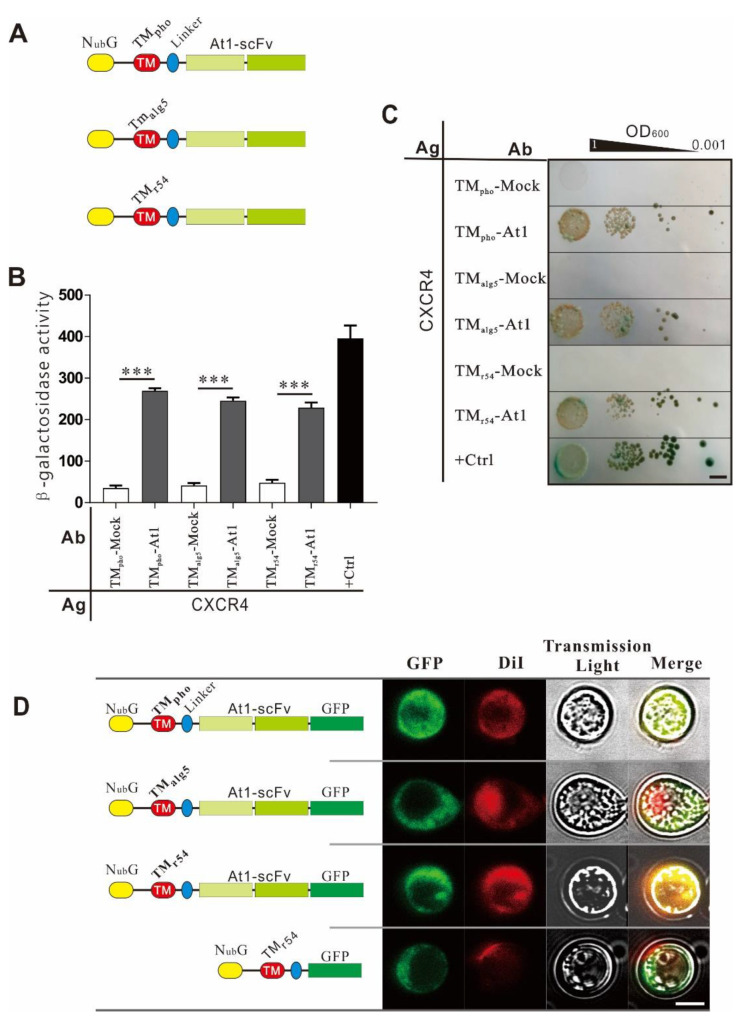
Optimization of transmembrane (TM) peptides. (**A**) Diagram of scFv modules with different TM peptides. Three transmembrane peptides (TM_Pho_, TM_alg5_, and TM_r54_) were fused between the N_ub_*G* domain and flexible linker (3×GGGGS), followed by At1-scFv in tandem. (**B**) β-Gal assay. The interaction strength between the antigen (CXCR4) module and the antibody (At1-scFv) module containing different TM peptides was measured using the β-galactosidase assay. The positive control was the same as in Figure 2E, the bar indicate 5 mm. (**C**) Cell growth assay. The growth of hybrid cells harboring antigen and antibody modules (same as (**B**)) were evaluated on the selection medium. (**D**) Subcellular localization of antibody modules with different TM peptides. pGBK-T7-At1-scFv-TM-N_ub_*G* was used as the backbone of antibody plasmids. TM_Pho_, TM_r54_, and TM_alg5_ were fused between the N_ub_*G* domain and linker (3×GGGGS), followed by At1-scFv and EGFP in tandem. pGBK-T7-EGFP-TM_Pho_-N_ub_*G* was used as a positive control. The antibody modules were expressed by Y187 and observed using laser scanning confocal microscopy. Images of EGFP, membrane-specific fluorescent dye DiI, and transmission light were captured and merged, the bar indicates 5 μm.

**Figure 4 ijms-22-04711-f004:**
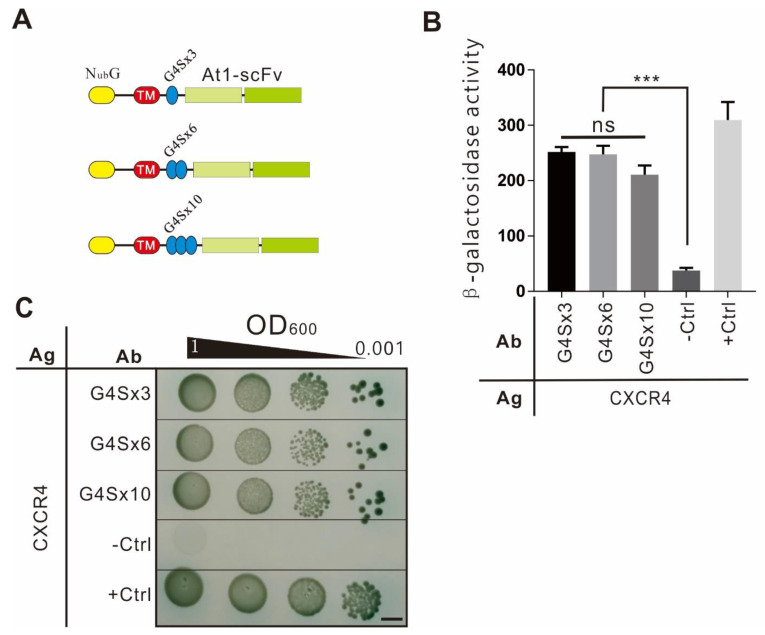
Optimization of flexible peptide length. (**A**) Diagram of scFv module with different lengths of flexible linkers. Flexible linkers composed of 3×GGGGS, 6×GGGGS, or 10×GGGGS were constructed into antibody modules, which were tested for their interaction strength with the antigen module using the following assays. (**B**) β-Gal assay. The interaction strength between CXCR4 and antibody modules containing variable lengths of the flexible linker was measured using the β-galactosidase assay. The positive and negative controls were the same as in Figure 2E. (**C**) Cell growth assay. The growth of hybrid cells harboring the antigen and antibody module as in (**B**) was evaluated by the selection medium. The bar indicates 5 mm.

**Figure 5 ijms-22-04711-f005:**
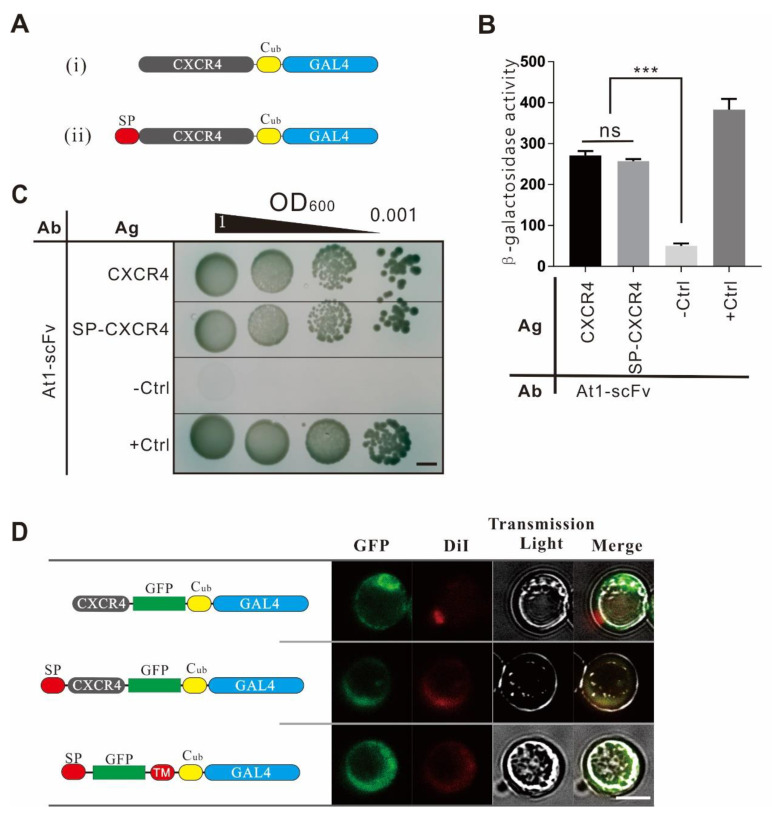
Optimization of antigen module localization by signal peptides. (**A**) Diagram of antigen structures w/o signal peptide (SP). (i) Human CXCR4 alone or (ii) CXCR4 led by a signal peptide (α-factor) were fused with C_ub_ and GAL4 in tandem forming antigen modules. (**B**) β-Galactosidase assay. Interaction strength between the two antigens and At1-scFv protein (*trans*-form) were measured using the β-galactosidase assay. TM_pho_-C_ub_-GAL4 × At1-scFv-N_ub_*G* was used as a negative control, and TM_pho_-C_ub_-GAL4 × At1-scFv-N_ub_*I* was used as a positive control. (**C**) Cell growth assay. Cell growth mediated by CXCR4/At1-scFv interaction was evaluated in the selection medium. The bar indicates 5 mm. (**D**) Subcellular localization of antigen module. The pGAD-T7-CXCR4-C_ub_-GAL4 vector was used as the backbone of antigen plasmids. EGFP was fused between the CXCR4 and C_ub_ domain, followed by GAL4 in tandem. pGAD-T7-EGFP-TM_Pho_-C_ub_-GAL4 was used as a positive control. The bar indicates 5 μm.

**Figure 6 ijms-22-04711-f006:**
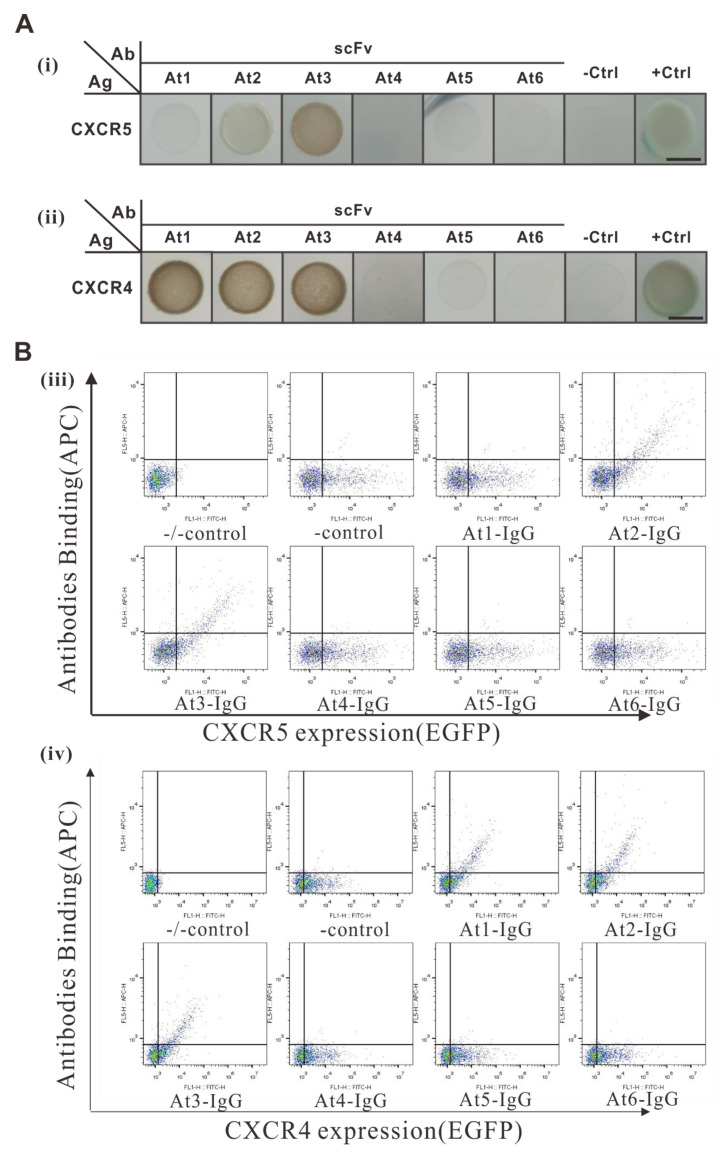
Antibody specificity detection in the AACD system. (**A**) Detection of specific antigen-antibody interaction by the AACD system. Six scFv, including At1 and At2 against CXCR4, At3 against CXCR5, and At4, At5, and At6 against other antigens, were tested for their specificity against CXCR4 (i) or CXCR5 (ii). CXCR4, CXCR5, and scFv sequences were cloned into the optimized AACD system, and the specific interactions were detected by cell growth assay. CXCR4-C_ub_-GAL4 and CXCR5-C_ub_-GAL4 × TM_pho_-N_ub_*G* were used as negative controls, and CXCR4-C_ub_-GAL4 and CXCR5-C_ub_-GAL4 × TM_pho_-N_ub_*I* were used as positive controls. The bars indicate 5 mm. (**B**) Detection of the antigen-antibody interaction using flow cytometry. As a corroboration, BHK21 cells expressing CXCR4 or CXCR5 were tested for their association with the aforementioned six scFv genes. BHK21 cells were stably transfected with CXCR4 or CXCR5 fused with EGFP. A recombinant IgG derived from the six scFv genes described above was purified and analyzed using flow cytometry. Antigen expression was detected by EGFP (green), and antibody binding was detected by an APC-labelled secondary antibody (red). −/−control: Parental BHK21 cells stained with isotype IgG; -control: antigen-expressing BHK21 cells stained with isotype IgG; 3 × 10^5^ cells were sampled for each cell line.

**Figure 7 ijms-22-04711-f007:**
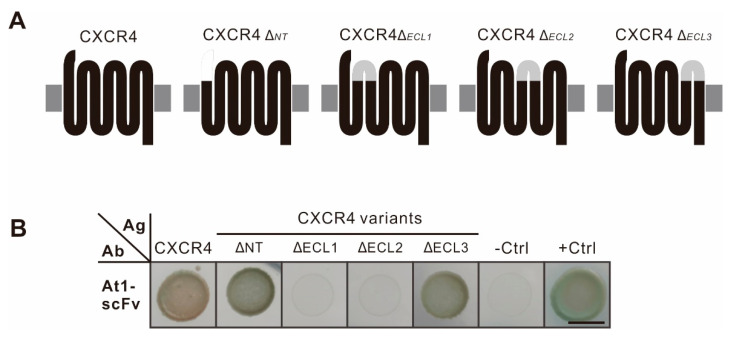
Determination of CXCR4 epitopes that mediate antigen-antibody interaction. (**A**) Diagram of CXCR4 mutants. Four extracellular fragments of CXCR4 were substituted with 2×GGGGS flexible peptides, named CXCR4_ΔNT_, CXCR4_ΔECL1_, CXCR4_ΔECL2_, and CXCR4_ΔECL3_, respectively. (**B**) Interaction assay. The interaction between mutant CXCR4 and its specific antibody, At1, was detected in the AACD system. The CXCR4 mutants were cloned into the antigen module, and their interaction with At1-scFv was detected. The interactions among CXCR4_ΔECL1_, CXCR4_ΔECL2_*,* and At1-scFv were hindered. The positive and negative controls were the same as in Figure 5C. The experiments were performed in triplicate. The bar indicates 5 mm.

## Data Availability

Not applicable.

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
