# Peer review of "Development of an Antigen-Antibody Co-Display System for Detecting Interaction of G-Protein-Coupled Receptors and Single-Chain Variable Fragments"

_ijms, 2021, doi:10.3390/ijms22094711_

Round 1
Reviewer 1 Report
The manuscript by Zhang et al. describes the development and optimization of a Y2H-based split ubiquitin system to detect interactions of scFvs with 2 selected GPCRs. The authors describe the optimization of several parameters within this system such as scFv topology, linker length and identity of the transmembrane domain.
Although the idea to use a Y2H system for detection and to study interactions of antibody fragments with their (hard to express) targets is not particularly novel, the description of the implementation of GPCR targets can be of use for people working in this field. GPCRs are notoriously hard to use in antibody discovery campaigns and while the authors mainly describe a proof-of-concept study, it should be possible to further develop this approach into an alternative to conventional biopanning strategies (as briefly mentioned in the Discussion). The scientific methodology used herein is robust and controls were included appropriately to confirm the validity of arguments. The results do support the conclusions made. Whether this approach can be further developed to be useful for library-based screening strategies remains to be determined but I do recommend publication of the manuscript with minor revision as detailed below:
General comments:
Despite being clear enough to be understood, the manuscript would greatly benefit from further proofreading with regards to proper use of the English language and scientific terminology. This is particularly apparent in the Introduction and Discussion section. I have indicated a few suggestions below but won’t be able to do this for the entire manuscript.
- What are off-targets in this context? I see this used in the abstract and discussion. Do the authors mean cross-reactivity?
- Page 1 line 41: hybrid tumor technique should be hybridoma technique
- Page 2 line 46 – the term recombining should be reserved for actual genetic recombination events. I would suggest to use ‘reconstitution’ when referring to solubilisation of membrane proteins in (artificial) lipid carriers.
- Figure references in the text (1Ai and 1Aii instead of 1A and 1B for example) need to be corrected.
- Line 90: I assume the authors mean selective plate instead of defective plate.
- Section 2.2: I think the authors should briefly describe the results of experiments shown in Figure 2D and 2E while referencing these figure panels. From the figure and caption it is obvious they came to the conclusion that trans-topological structure is able to activate expression of the reporter gene while the cis-topology did not but it would be helpful to have this described in the main text in a bit more detail as well.
- Line 275: I suggest to begin the sentence: To optimise the system,…
- Lines 280-283: The sentence lacks an appropriate scientific description of conformational changes and their effects. Bending backwards is highly dependent on one’s definition of what is front and what is back. This is usually hard to define within protein structures and I would limit this discussion to the fact the scFv might have to undergo a larger, potentially unfavorable conformational changes paired with the problem of steric hinderance when used in this topology. Varying affinities of scFvs with their target antigens depending on topologies and orientations is definitely not unheard of which is, I suspect, the reason why the authors chose to test cis and trans topologies in the first place. For the sake of clarity, I also suggest to either delete or rephrase the sentence that talks about introducing uncertainty into the system. Uncertainty usually refers to statistical descriptions which used in this context would be inaccurate.
- Line 287: The transmembrane peptide does not express the scFv but anchors or localises it to (or outside for that matter) the membrane.
- Line 289: What exactly is the innate advantage of yeast transmembrane proteins? Their ability to not aggregate or be engaged in non-specific interactions? Can the authors please cite references for this argument? I am not a yeast expert but I’m pretty sure that there are yeast membrane proteins (receptors etc.) that undergo transmembrane domain-mediated oligomerization that, if chosen, might pose a problem with respect to non-specific interactions in this system. Conversely, transmembrane proteins in other species exist that may be equally suited for this approach. I’m not saying the choice of transmembrane domains for this system was inadequate but I’d ask the authors to be a bit more specific here while supporting their arguments with references.
- Line 334: What is an ideal antibody library? Antibody library alone in this context is sufficient.
- Line 344: should be: which decreases the risk …
- Please use the correct citation style to reference patents in the bibliography
Author Response
Dear reviewer,
We greatly appreciate your careful review and thank you for your constructive suggestions, which significantly improved our manuscript.
According to your comments, we made the following revisions or improvements. (Italic words represent the comments given by reviewers)
Reviewers' comments and author’s responses::
What are off-targets in this context? I see this used in the abstract and discussion. Do the authors mean cross-reactivity?
Response: Yes, “off-targets” means “cross-reactivity”. We changed off-targets to cross-reactivity.
Page 1 line 41: hybrid tumor technique should be hybridoma technique
Response: “hybrid tumor technique” was changed to “hybridoma technique”.
Page 2 line 46 – the term recombining should be reserved for actual genetic recombination events. I would suggest to use ‘reconstitution’ when referring to solubilisation of membrane proteins in (artificial) lipid carriers.
Response: “recombining” was changed to “reconstitution”.
Figure references in the text (1Ai and 1Aii instead of 1A and 1B for example) need to be corrected.
Response: Page 2 line 72 “1Ai” was changed to “1A”.
Page 2 line 79 “1Aii” was changed to “1B”.
Page 2 Line 90: I assume the authors mean selective plate instead of defective plate.
Response: “defective plate” was changed to “selective plate”.
Section 2.2: I think the authors should briefly describe the results of experiments shown in Figure 2D and 2E while referencing these figure panels. From the figure and caption it is obvious they came to the conclusion that trans-topological structure is able to activate expression of the reporter gene while the cis-topology did not but it would be helpful to have this described in the main text in a bit more detail as well.
Response:
“The results showed that the antibody module with the trans-topological structure significantly activated the reporter gene, while the cis structure did not.” It was revised:
The results showed that the yeast with antibody module of trans-topological structure and CXCR4-Cub-GAL4 grew better in the selective plate and had higher β-Gal activity than the cells with the cis-topology. It demonstrated that the trans-topology significantly activated the reporter gene, while the cis structure did not.
Line 275: I suggest to begin the sentence: To optimise the system,…
Response: “polish” was changed to “optimize”.
Lines 280-283: The sentence lacks an appropriate scientific description of conformational changes and their effects. Bending backwards is highly dependent on one’s definition of what is front and what is back. This is usually hard to define within protein structures and I would limit this discussion to the fact the scFv might have to undergo a larger, potentially unfavorable conformational changes paired with the problem of steric hinderance when used in this topology. Varying affinities of scFvs with their target antigens depending on topologies and orientations is definitely not unheard of which is, I suspect, the reason why the authors chose to test cis and trans topologies in the first place. For the sake of clarity, I also suggest to either delete or rephrase the sentence that talks about introducing uncertainty into the system. Uncertainty usually refers to statistical descriptions which used in this context would be inaccurate.
Response: We agree with the reviewer’s opinion. We revised the sentence of “where scFv needs to fold backward to make the antigen-binding region accessible to the membrane. This structure introduced some uncertainty into the system” to “where scFv might have to undergo a larger, potentially unfavorable conformational changes paired with the problem of steric hinderance when used in this topology.”
Line 287: The transmembrane peptide does not express the scFv but anchors or localises it to (or outside for that matter) the membrane.
Response: “express” was change to “anchors”.
Line 289: What exactly is the innate advantage of yeast transmembrane proteins? Their ability to not aggregate or be engaged in non-specific interactions? Can the authors please cite references for this argument? I am not a yeast expert but I’m pretty sure that there are yeast membrane proteins (receptors etc.) that undergo transmembrane domain-mediated oligomerization that, if chosen, might pose a problem with respect to non-specific interactions in this system. Conversely, transmembrane proteins in other species exist that may be equally suited for this approach. I’m not saying the choice of transmembrane domains for this system was inadequate but I’d ask the authors to be a bit more specific here while supporting their arguments with references.
Response: In choosing the right transmembrane peptide we tested three candidates, two from the yeast proteins and one from human. There were no obvious differences among them in their transmembrane localization and reporter gene activation (Fig. 3). We chose TMpho transmembrane peptide from the yeast protein Pho90 in constructing our AACD system. However, this result-based choice has no support of publications. Therefore, we deleted the sentence in line 289: “Typically, the secreted transmembrane proteins of yeast have an innate advantage no matter how they are synthesised, expressed, or anchored on the cell membrane, whereas transmembrane proteins from other species may not have this advantage. Nevertheless, the transmembrane proteins secreted by yeast might bind to the proteins on the yeast cell membrane in a non-specific way.”
Line 334: What is an ideal antibody library? Antibody library alone in this context is sufficient.
Response: The word “ideal” was deleted.
Line 344: should be: which decreases the risk …
Response: The word “lows” was change to “decreases”.
Please use the correct citation style to reference patents in the bibliography.
Response: We corrected the references of number9, 13, 14 accordingly.
Yours sincerely,
Dr. Jingjing Li
School of Pharmacy,
Shanghai Jiao Tong University
Email: lijj@sjtu.edu.cn
Tel: (86)-21-34205769
Reviewer 2 Report
The topic of the paper is novel and original. The paper itself is well structured and organized. Experiments were performed meticulously. The authors proved that the antibody works in a yeast system. My concern is whether it works in a mammalian cell model. Experiments demonstrating that the CXCR4 antibody cam be correctly anchored in a mammalian cell membrane would bring a plus to the paper. Would the system work for another type of transmembrane protein? For example another GPCR protein.
Author Response
Dear Reviewer,
We greatly appreciate your careful review and thank you for your constructive suggestions, which significantly improved our manuscript.
According to your comments, we made the following revisions or improvements. (Italic words represent the comments given by reviewers)
Reviewer’s comments:
The topic of the paper is novel and original. The paper itself is well structured and organized. Experiments were performed meticulously. The authors proved that the antibody works in a yeast system. My concern is whether it works in a mammalian cell model. Experiments demonstrating that the CXCR4 antibody cam be correctly anchored in a mammalian cell membrane would bring a plus to the paper. Would the system work for another type of transmembrane protein? For example another GPCR protein.
Authors’ response:
We did not perform similar experiments on mammalian cells. However, we successfully tested another GPCR and its interaction with its antibody, which is CXCR5 (Fig. 6). We will examine additional GPCRs to further validate the versatility of this AACD system. .
Yours sincerely,
Dr. Jingjing Li
School of Pharmacy,
Shanghai Jiao Tong University
Email: lijj@sjtu.edu.cn
Tel: (86)-21-34205769